# Precision Detection of Real-Time Conditions of Dairy Cows Using an Advanced Artificial Intelligence Hub

Kim Margarette Corpuz Nogoy [1,2], Jihwan Park [1], Sun-il Chon [1], Saraswathi Sivamani [1], Min-Jeong Park [1], Ju-Phil Cho [3], Hyoung Ki Hong [2], Dong-Hoon Lee [4,*] and Seong Ho Choi [2,*]

[1] ThinkforBL Consultancy Services, Seoul 06236, Korea; margarette@thinkforbl.com (K.M.C.N.); jihwan.park@thinkforbl.com (J.P.); sichon@thinkforbl.com (S.-i.C.); sara@thinkforbl.com (S.S.); mjpark@thinkforbl.com (M.-J.P.)

[2] Department of Animal Science, Chungbuk National University, Cheongju City 28644, Korea; hhkdoctor@hanmail.net

[3] School of IT Information and Control Engineering, Kunsan National University, Gunsan City 54150, Korea; stefano@kunsan.ac.kr

[4] Department of Biosystems Engineering, Chungbuk National University, Cheongju City 28644, Korea

* Correspondence: leedh@cbnu.ac.kr (D.-H.L.); seongho@cbnu.ac.kr (S.H.C.); Tel.: +82-43-261-2579 (D.-H.L.); +82-43-261-2544 (S.H.C.); Fax: +82-43-261-2579 (D.-H.L.); +82-43-261-2544 (S.H.C.)

**Featured Application: This study demonstrates an artificial intelligence-based tool that could identify the real-time condition of dairy cows by using underutilized data generated from the Internet of Things equipment in dairy farms.**

**Abstract:** One of the main challenges in the adoption of artificial intelligence-based tools, such as integrated decision support systems, is the complexities of their application. This study aimed to define the relevant parameters that can be used as indicators for real-time detection of heat stress and subclinical mastitis in dairy cows. Moreover, this study aimed to demonstrate the use of a developed data-mining hub as an artificial intelligence-based tool that integrates the defined relevant information (parameters or traits) in accurately identifying the condition of the cow. A comprehensive theoretical framework of the data-mining hub is demonstrated, the selection of the parameters that were used for the data-mining hub is listed, and the relevance of the traits is discussed. The practical application of the data-mining hub has shown that using 21 parameters instead of 13 and 8 parameters resulted in a high overall accuracy of detecting heat stress and subclinical mastitis in dairy cows with a high precision effect reflecting a low percentage of misclassifying the conditions of the dairy cows. This study has developed an innovative approach in which combined information from different independent data was used to accurately detect the health and wellness status of the dairy cows. It can also be implied that an artificial intelligence-based tool such as the proposed theoretical data-mining hub of dairy cows could maximize the use of continuously generated and underutilized data in farms, thus ultimately simplifying repetitive and difficult decision-making tasks in dairy farming.

**Keywords:** dairy farming; precision and accuracy; data mining; heat stress; subclinical mastitis; artificial intelligence

## 1. Introduction

Dairy farming is a decision-intensive field of agriculture that must rely on a holistic system approach accounting for the well-being, physiological, behavioral, and health conditions of the animal to meet the specific requirements of the dairy cows. Its management requires cooperation and close communication between dairy experts, farm managers, and dairy factories. The birth of technological innovations, such as the use of the Internet of Things, has helped in detecting and identifying the condition of the cows and has contributed to the betterment of the welfare of the animals. State-of-the-art measures in

dairy farming that enable feed intake measurements include acoustic monitoring [1–3], chewing activity, or jaw movement augmented with sensor data [4], noseband pressure sensors [5], and electromyography [6]. In automated health diagnostics, which are commercially and widely available [7,8], the movement, behavior, and physiological conditions of dairy cows are used to relay information about their well-being and health, such as MooMonitor [9] and Zigbee [10]. The use of these advanced monitoring and diagnostic tools, such as applications and expert systems, are normally restricted between the user and the proprietary of the tool, and these applications mostly focus on one problem and do not involve data integration [11]. Analyzing independent and disparate data could be informative and could create isolated judgments about the animals, and hence, drive the dairy farmers in making decisions based on intuition and experiences rather than based on the unique requirements of the cows. A revolution in the era of artificial intelligence and big data has tried to resolve this by making use of the massive volumes of widely varied data that can be taken, analyzed, and interpreted [12], and integrated to produce a more insightful outcome. Some of the dairy farm decision system tools include, but are not limited to, the Dairy Brain [13], My Dairy Dashboard [14], and Cargill Dairy Entelligen^TM [15]. However, the practical applications and adoption of these efforts are not sustained in dairy farms, and in fact, only 22 peer-reviewed papers have reported data integration in decision making in dairy farming [16]. Taking full advantage of the opportunities of the created data has been difficult for dairy farmers and animal scientists alike to accurately identify the condition of the cows for efficiently maximizing the productivity of the cows.

Hence, this study aimed to show the possibility of developing an artificial-based tool that integrates relevant information (parameters or traits) needed for accurately identifying the condition or performance of the cow. Specifically, the objectives of the study were (1) to demonstrate a comprehensive theoretical framework of the use of a data-mining hub in improving the precision of identifying the condition of dairy cows; (2) to provide a list of parameters or traits of the dairy cows that are relevant in detecting conditions of cows, and (3) to conduct a practical application to compare the number of traits or parameters needed to identify the statuses of the specific cow, which in this study, are identified as a normal condition, heat-stress, and subclinical mastitis risk (SCMR).

## 2. Materials and Methods

### 2.1. Data Collection

The data of this study consisted of the online and offline farm data collected from a commercial dairy farm in South Korea. Offline farm data pertains to the herd records, such as the animal type (heifer, pregnant, calving, lactating, dry-off), breed type (Holstein-Friesian, Jersey, Guernsey), breeding system (straightbred, crossbred), parity (1–7), and lactation stage (early, mid, late). Online farm data pertains to the records that are measured daily, transferred to the server of the farm, and collected for processing using the proposed data-mining hub. These online farm data include the environmental conditions measured by the meteorological equipment fitted in the farm, the physiological conditions of the dairy cows as measured by the sensors equipped in the farm, the behavioral data of the dairy cows as measured by sensors or surveillance video cameras, the milk yield and composition records as analyzed by a commercial milking laboratory connected to the farm, and lastly the feed information, including the composition of the rations, feed intake of the cow per day, and the cost per kilogram of the ration as measured by the automatic feeding system in the farm. The set of equipment used by the farm in measuring parameters included the following: body weight measured by walk-over weighing system; body condition score (BCS) of the cows scored manually by farm staff; body temperature measured by thermosensors; rumen temperature, rumen pH, and rumination measured by rumen boluses; respiration rates measured by heart rate monitors, standing, lying, sitting, and drinking; behaviors of the cows measured by pedometers and surveillance video cameras. All of the unprocessed data collected on the farm were made available by the dairy farmer and provided to the research group for the perusal of this study.

## 2.2. Data Selection

It was anticipated that the unprocessed data collected by various equipment in farms generated disproportionate values that need processing to transform into usable numbers. Heterogeneities of the data, such as the system of the data source, the sensors and devices used in measuring physiological and behavioral values, and the formats of the data, were processed by the trained members of the group. Data mining and data-processing tool scripts were used to detect and remove inaccurate, duplicated, and corrupted data. The cleaned and harmonized data were loaded to the proposed data-mining hub, which followed the format shown in Figure 1, but not all the data were used. The format followed a standardized sampling scheme where the cleaned and harmonized data for every day was encoded per row according to the parameters they belonged to. The animal information and parameters were sectioned per row, while the data per day was encoded per column. The information of the dairy cows was interconnected by the unique identity traceability number of the dairy cow. The datasets of three individual cows (triplicates) were mapped and grouped through the traceability number of the cow. The selected individual cows were all fourth parity lactating straightbred Holstein cows. In each individual cow identified by its traceability number, processed data under different parameters generated were further selected relevant to this study. For instance, environmental conditions, such as ambient temperature and relative humidity, were selected as climate records of the cow while wind speed and sun exposure were only stored in the data-mining hub; records about feed, such as cost of ration and ration composition, were saved and stored while the feed intake dataset was selected for the study. A total of twenty-one (21) parameters were used in detecting the status of the cows, and the datasets were grouped into three groups: the HI, MID, and LO groups. The HI group used 21 parameters, while MID and LO groups used thirteen (13) and eight (8) parameters, respectively. The MID group (13 parameters) consisted of the four behaviors, seven milk performance indexes, and the feed intake. The LO group (8 parameters) consisted of the seven milk performance indexes and the feed intake.

| | Day 1 | Day 2 | Day 3 | Day 4 | Day 5 | Day 6 | Day 7 | Day 8 | Day 9 | Day 10 |
|---|---|---|---|---|---|---|---|---|---|---|
| Animal unique traceability number | 123 | 123 | 123 | 123 | 123 | 123 | 123 | 123 | 123 | 123 |
| Animal information | Record | Record | Record | Record | Record | Record | Record | Record | Record | Record |
| Environmental Condition | Data | Data | Data | No data | Data | Data | Data | No data | No data | No data |
| Physiological Values | No data | No data | Data | Data | Data | No data | No data | Data | Data | Data |
| Behavioral Values | Data | Data | Data | No data | Data | Data | Data | Data | Data | Data |
| Milk Production | Data | Data | Data | Data | Data | Data | Data | Data | Data | No data |
| Milk Composition | Data | Data | Data | Data | Data | Data | No data | Data | Data | Data |
| Feed Records | Data | Data | Data | No data | Data | Data | Data | Data | Data | Data |

**Figure 1.** The basic format of the data-mining hub in which the cleaned and harmonized data were loaded. The figure illustrates the scheme of data collection and integration format used in the study based on the report of [17].

## 2.3. Selection of Parameters to Be Used in the Data-Mining Hub

For this study, parameters that directly and indirectly affect heat stress and subclinical mastitis in dairy cows were focused on. The parameters used for the development of the data-mining hub were derived from the five main conditions of the cows: (1) environmental conditions, (2) physiological conditions, (3) behavioral conditions, (4) milk performance, and (5) feed intake, as specified in Table 1.

*Environmental conditions* include the ambient temperature (Ta), relative humidity (RH), and temperature-relative humidity index (THI). The THI is composed of the effects of Ta and RH and is used as an important environmental index to assess the state of heat stress and the degree of heat stress in animals. For instance, according to [18], a THI value less than 72 will give an optimal condition for dairy cows, whereas THI ranging from 73 to 79, 80 to 89, and THI greater than 90 will give dairy cows mild, moderate, and severe heat stress,

respectively. In the same way, it was reported that subclinical and clinical mastitis risks in dairy cows are increased when exposed to a hot environment [19,20]. Therefore, high Ta, RH, and THI should reflect to increase heat stress and subclinical mastitis conditions in the data-mining hub.

*Physiological conditions* include body weight (BW), body condition score (BCS), body temperature, rumen temperature ($T_{rum}$), rumen pH, and respiration rate. As far as the research of this study, the body weight and BCS of dairy cows have no direct relation with heat stress conditions. In subclinical mastitis, it was also found that BCS has no significant relation with subclinical mastitis, but it has been reported that the bodyweight of the dairy cows is positively related to somatic cell count (SCC) in milk and the somatic cell count is a well-established indicator of subclinical mastitis [21]. Body temperature gives a direct measure of the heat stress level of individual dairy cows [18] and $T_{rum}$ normally increases with increased ambient temperature [22]. It has also been reported that a high rise in body temperature [23] and $T_{rum}$ [24] are accompanied by subclinical mastitis. Rumen pH is reduced in heat-stressed cows [25] and also tends to decrease in dairy cows that will be affected by subclinical mastitis [26]. Dairy cows affected with subclinical mastitis have been reported to increase respiratory rate [27] while dairy cows that are experiencing milk heat stress have been reported to show 61 breaths per minute [28] and 120 breaths per minute for severe heat stress [29]. The physiological regulation in dairy cows is the most critical factor in their productive capacity. Therefore, the combined physiological parameters are an eminent factor for accurate detection of heat stress, subclinical mastitis, and normal conditions of the cows using the data-mining hub.

*Behavioral conditions* in the data-mining hub include ruminating, standing, lying/sitting, and drinking, which are important indexes to measure the health and welfare of dairy cows. The ruminating time of dairy cows is increased when the THI is increased [30,31], whereas a reduction in the ruminating time of dairy cows with subclinical mastitis has been reported [32]. The standing behavior of the dairy cows was reported to be prolonged when heat-stressed [33] and when affected with subclinical mastitis [32]. Consequently, decreases in the lying time of the dairy cows were observed in animals affected with subclinical mastitis [24,32]. Heat-stressed dairy cows, on the other hand, were observed to be restless [34]. Heat-stressed dairy cows also increased their drinking behavior [35] while dairy cows affected with subclinical mastitis reduced their water intake as evidenced by a reduction in drinking behavior [32]. These subtle modifications in the behavior of the dairy cows materialize and can be observed before the clinical signs of the disease and, therefore, can be helpful in the accurate prediction of the heat stress, subclinical mastitis, or normal condition of the animals.

*Milk performance indexes* selected for the use in the data-mining hub include the milk yield and the composition, such as somatic cell count (SCC), milk solids-not-fat (SNF), milk fat (MF), milk protein (MP), milk urea nitrogen (MUN), and milk lactose. Heat stress tends to increase the SCC in milk, especially during hot months [36,37] leading to the reduction of the quality of the milk produced by the dairy cows. Heat stress has also been reported to reduce the SNF [38], MF, MP, and milk lactose [36,38–41] and tends to increase MUN [42] in dairy cows. Milk yield decreased two days after the initiation of heat stress [43] and is also reduced in dairy cows with subclinical mastitis [44]. Milk composition of dairy cows with subclinical mastitis such as SNF, MF, MP [45] is reduced, while the MUN has been observed to tend to decrease [46]. Most importantly, the SCC of milk with subclinical mastitis expresses a sharp increase in count [24,47]. Specifically, it has been reported that subclinically infected dairy cows with mastitis have shown a somatic cell number of above $2.0 \times 10^5$ per mL of milk [48]. This increase in somatic cell count has shown a negative relationship with milk lactose [48–51] thus, subclinical mastitis-infected dairy cows could reduce milk lactose. The changes in the milk composition are associated with the metabolic status and health of dairy cows, and therefore, the combined effect of these milk performance parameters could help predict the conditions of the animals more accurately.

*Feed intake* is the most important index in measuring the health and wellness of dairy cows as the changes in intake could be measured as early as two weeks before calving or any clinical problems [52] and thus, this parameter could help in identifying cows that are at risk. Feed intake of cows that are heat-stressed is reduced as early as one day before the initiation of the heat stress [43] and is normally reduced in dairy cows with subclinical mastitis [53].

The 21 parameters selected and used in developing the data-mining hub that will detect the conditions of the dairy cows should reflect the heat stress, subclinical mastitis, or normal status of the cows more accurately than a single or few parameters alone.

**Table 1.** Selected parameters that were used for detecting the status of the cows.

| Category | Parameters | Heat Stress | Subclinical Mastitis (SCM) |
|---|---|---|---|
| Environmental conditions | Ambient Temperature (Ta) | Increased temperature tends to increase risk for heat stress | Increased temperature tends to increase risk for SCM |
| | Relative Humidity (RH) | Increased humidity increases THI | Increased humidity increases THI |
| | Temperature-Humidity Index (THI) | <72 THI: Normal condition; 73–79 THI: Mild heat stress; 80 to 89: Moderate heat stress >90: Severe heat stress [18] | >79 THI values are associated with a higher risk of CM development [20] |
| Physiological body conditions | Bodyweight (BW) | No changes were reported in the bodyweight of the cows as affected by heat stress | Bodyweight is positively related to SCC [21] |
| | Body condition score (BCS) | No changes were reported in the bodyweight of the cows as affected by heat stress | BCS is not significantly related to SCM [21] |
| | Body temperature | Heat stress dairy cows show increased body temperature [18] | SCM is accompanied by a high rise in temperature [23] |
| | Rumen temperature | Rumen temperature increased with increasing environment temperature [22] | Cows affected with SCM increased rumen temperature [24] |
| | Rumen pH | Heat stress results to reduced pH [25] | Low rumen pH tends to be associated with subclinical mastitis [26] |
| | Respiration rate | Cows are heat-stressed at 61 breaths per minute [28] and severe heat stress at 120 breaths per minute [29] | Cows increased respiratory rate if affected with moderate mastitis [27] |
| Behavior status | Ruminating | Rumination time decreases with increasing THI [30,31] | Cows reduce ruminating time when affected [32] |
| | Standing | Prolonged standing indicates heat stress to cows [33] | Cows increased standing time when affected [32] |
| | Lying | Heat-stressed cows show a restless state [34] | Cows decreased lying time when affected [24,32] |
| | Drinking | Heat-stressed cows require increased drinking water [35] | Cows drink less when affected [32] |
| Milk performance indexes | Somatic cell count (SCC) | SCC tend to rise with temperature and humidity, especially during the summer months [37] | SCM show sharp increase of SCC [24,47] |
| | Milk solids-not-fat (SNF) | Heat stress reduced SNF [38] | SCM reduced SNF [45] |
| | Milk fat (MF) | Heat stress reduced MF [40] | SCM reduced MF [45] |
| | Milk protein (MP) | Heat stress decreased MP [41] | SCM reduced MP [45] |
| | Milk urea nitrogen (MUN) | Heat stress tends to increase MUN 42 | SCM tend to reduce MUN [46] |

**Table 1.** *Cont.*

| Category | Parameters | Heat Stress | Subclinical Mastitis (SCM) |
|---|---|---|---|
| Milk performance indexes | Milk lactose (ML) | Heat stress reduced ML [40] | SCM reduced ML [45,48–51,54] |
| | Milk yield (MY) | Milk yield decreased 2 days after initiation of heat stress [43] | SCM reduced milk yield [44] |
| Feed | Feed intake (FI) | Feed intake decreased 1 day after initiation of heat stress [43] | SCMreduced feed intake [53] |

### 2.4. K-Nearest Neighbor Algorithm and Statistical Analysis

The machine learning algorithm k-nearest neighbor (KNN) was used for the classification and identification of the status of the cows. The k-nearest neighbor algorithm is a classical non-parametric method classification algorithm. In a KNN classifier, the model is not trained, but all instances are retained. The class of a new input is concluded by finding the k-closest adjacent training instances and computing the class connection of the training inputs. The new input is deduced as the class with the most vote among the k-nearest training instances in which *k* is a positive integer and is typically small. The main idea is if the majority of the k-closest values of the input samples belong to a specific category or class, then the new samples with closely similar values will belong to the same category. The KNN is easily implemented and suitable for multi-classification problems.

For the classification process, the cross-validation method of leave-one-out was used, meaning that the data of the three individual cows were used for the training set, and another set of data from another individual cow was used for validating the set. The evaluation for the effect of classification was based on the accuracy, Kappa coefficient, and precision. The accuracy and precision were calculated as follows: (1) accuracy = $(TP + TN)/(P + N)$, and (2) precision = $TP/(TP + FP)$; where: TP as true positive is the number of positive samples correctly classified as positive samples, TN as true negative is the number of negative samples correctly classified as negative samples, FP as false positive is the number of negative samples incorrectly classified as positive samples, P as the number of positive samples, and N as the number of negative samples. The Kappa coefficient is a calculation of classification accuracy. Kappa coefficient between 0 to 0.2 represents a very low level of consistency, 0.41 to 0.60 represents medium consistency, 0.61 to 0.80 represents a high degree of consistency, and greater than 0.80 represents complete consistency to the true value [54].

The data preprocessing, preparation, and algorithm programming after the acquisition of data were conducted using python 3.7.0 [55]. The commonly used scientific computing toolkits, such as matplotlib and scikit-learn, were also based on python 3.7.0.

The Pearson correlation between the parameters and the conditions of dairy cows was analyzed using the scientific plotting tool in python. The results of the practical application were analyzed by using the general linear model of SAS, and if differences were detected, data were further analyzed using Duncan Multiple Range Test (DMRT). The statistical significance level for all comparisons of the groups was set to 0.05 if not otherwise indicated.

### 3. Results and Discussion

#### 3.1. Developed ThinkDairy Data-Mining Hub for Detecting Conditions of Dairy Cows: A Theoretical Framework

The proposed ThinkDairy data-mining hub (Figure 2) showed how massive data collected from the dairy farms could be utilized in detecting conditions of dairy cows to support the creation of decisions for farmers. It is designed to carry out real-time assistance to farmers by cleaning and integrating data and be used for running multiple analyses that consider numerous relevant factors to improve the precision and accuracy of detection

of the health and wellness of the animals and ultimately, provide a precise calculation of feeding rations to individual cows according to their specific needs. The data-mining hub consisted of offline and online farm data. Offline farm data pertains to the animal records such as the animal type (heifer, pregnant, calving, lactating, dry-off), breed type (Holstein-Friesian, Jersey, Guernsey), breeding system (straightbred, crossbred), age in months, days in the cycle, days pregnant, days since calving, calving interval, parity (1–7), and lactation stage (early, mid, late). These offline records are used for the calculation of feed rations to dairy cows based on their real-time condition. However, this study focuses on the detection of the real-time condition of the cows only, and the application of the calculation of feed ration to dairy cows will be used for further studies. This section was stated for the sake of giving an overview of the flow of function of the data-mining hub. Online farm data pertains to the records that are measured daily, transferred to the server of the farm, and collected for processing using the proposed data-mining hub. The data collected includes (1) environmental condition: ambient temperature, relative humidity, temperature-humidity index, and wind and sun exposure; (2) physiological status such as body weight, body condition score (BCS), temperature, rumen conditions, breathing, and heart rate; (3) behavioral status: eating, standing, rumination, walking behaviors; (4) milk yield and composition records such as somatic cell count, solids-not-fat (SNF), milk protein, milk fat, milk lactose, protein to fat ratio, and milk urea nitrogen; and (5) feed records. These offline and online data are continuously collected from farms, stored in servers, and go through the customary data warehouse process called ETL, which is extraction, transformation, and loading. The assigned computer contains the files of the farm into specific directory structures determined by, i.e., farm source, data source, and date of acquisition. On the arrival of a new set of data files, the staging area or framework identifies the data files based on the directory structure and implements a predetermined code of one or more programs to extract the information from those files and validate it. Validations during extraction reconcile the records with the source data, separate the source data from unwanted loaded data, check the data type, and clean the source data from duplicates and fragmented data. The process of transforming the data begins after cleaning the extracted real-time data from the source and by data mapping, where a set of functions such as language-processing tool scripts are used to harmonize and standardize the extracted data. The standardized data are then integrated, collated, and utilized by a common format and structure or keys; for instance, in this study, it is the animal traceability identification number. The cow traceability identification number links all relevant data sources regarding the specific individual cow, regardless of the differences in the data source or farms. The standardized data are loaded thereafter into the data-processing hub named ThinkDairy Hub in a table format, as shown in Figure 1. These offline and online data will then be run using classifier algorithms to identify the real-time condition of the cows and will create an animal performance score. The animal performance score will be the basis for the recommended action and feed rations to farmers. For instance, in dairy farms, cows are identified by a unique traceability identity number. In order to be able to compare all records concerning a specific cow, the offline and online records using the traceability identity number are selected in order to analyze and compare all records for the specific cow. Offline and real-time data from the sensors and cameras are identified by the unique traceability identity number, extracted and transformed into a usable data form, and relevant data is finally loaded into a table view as shown in the blue, yellow, and black table in Figure 2. The 21 relevant parameters such as the temperature, humidity, THI, body weight, BCS, body temperature, rumen temperature, rumen pH, respiration rate, ruminating, standing, resting, drinking, milk composition and milk yield, and feed intake were then classified by the k-nearest neighbor classifier to produce the animal performance score (orange table). According to the performance score of the specific cow, whether it is a numerical or verbal score, an action measure is displayed for the farmers. At the same time, if the suggested action identifies increase, decrease, or maintain feed ration, the data-processing hub will run modeling and simulation to compute feed rations specific

for the requirements of the cow (green table). The part of modeling and simulation for the action and feed rations are only stated in this manuscript for the sake of discussing the overview of the whole function of the data-mining hub. In this study, the identification of the real-time condition, specifically heat stress and subclinical mastitis of the dairy cows, are further discussed. If successfully integrated, these offline and online data can generate informative and accurate interpretations rather than giving disparate conclusions when analyzed independently. With the unceasing growth of the use of sensors and new and different technologies in dairy farms [13,56], as well as in different industrial fields [57,58], the ThinkDairy data-processing hub would be a useful tool to continuously access, stream, and integrate data information from dairy farms. The application of the proposed developed ThinkDairy data-mining hub is further discussed in the following sections of the manuscript.

### 3.2. Relationship of the Parameters with Heat Stress and Subclinical Mastitis

The parameters that directly and indirectly affect heat stress and subclinical mastitis in dairy cows were focused on in this study. The 21 parameters selected and used in developing the data-mining hub that detect the conditions of the dairy cows reflected the heat stress, subclinical mastitis, or normal status of the cows more accurately than a single or few parameters alone. As shown in Figure 3, the correlation of temperature and temperature-humidity index (THI) to health conditions ranged from 0.75 to 1.00, while relative humidity (RH) scored a −0.75 correlation. Other parameters that have shown a positive correlation to health conditions are respiration rate and milk urea nitrogen (correlation = 0.75–1.00) and standing, drinking, and somatic cell count (correlation = 0.25–0.50). The parameters that have shown a high negative correlation to health conditions were body weight, rumen pH, rumination, lying/sitting, milk yield, and feed intake at −0.75–1.00 correlation, while milk lactose, milk protein, milk fat, milk solids nonfat (SNF), rumen temperature, body temperature, and body weight showed a slightly lower negative correlation at −0.25–0.50. When dairy cows are affected with heat stress or subclinical mastitis, physiological body conditions of the animals (body weight, body temperature, rumen temperature, and rumen pH) were reduced. The bodyweight reduction is expected due to reduced feed intake when dairy cows are experiencing discomfort due to heat stress or when cows are affected with subclinical mastitis. Additionally, rumination of the dairy cows was reduced when cows were affected by heat stress and subclinical mastitis. The reduced rumination could have affected the reduction of rumen temperature and pH. Interestingly, the resting behavior (lying or sitting) of the dairy cows declined when affected by heat stress or subclinical mastitis. Normally, dairy cows affected with illnesses or diseases decrease their moving behaviors and increase their resting behavior. The discomfort feeling of the dairy cows due to heat stress or subclinical mastitis could have made the animals restless, thus increasing their standing behavior (correlation = 0.50). In terms of milk performance indexes, SNF, milk fat, milk protein, milk lactose, and milk yield declined when dairy cows were affected with heat stress or subclinical mastitis. This result signifies that low-quality milk could diagnose dairy cows. The real-time measurement of milk performance could be a good indicator of heat stress or subclinical mastitis in dairy cows. The positive correlations of the respiration rate, standing and drinking behaviors, and somatic cell count with the health condition of the dairy cows suggest that the welfare of dairy cows are negatively affected, as evidenced by its increased panting and restlessness, thus affecting the microorganism load in the milk. The findings of the correlation suggest that the parameters selected for the development of the data-mining hub in accurately detecting the health condition of dairy cows can be used as indicators of heat stress and subclinical mastitis. The selected parameters used in the data-mining hub have proven relevant in health condition detection.

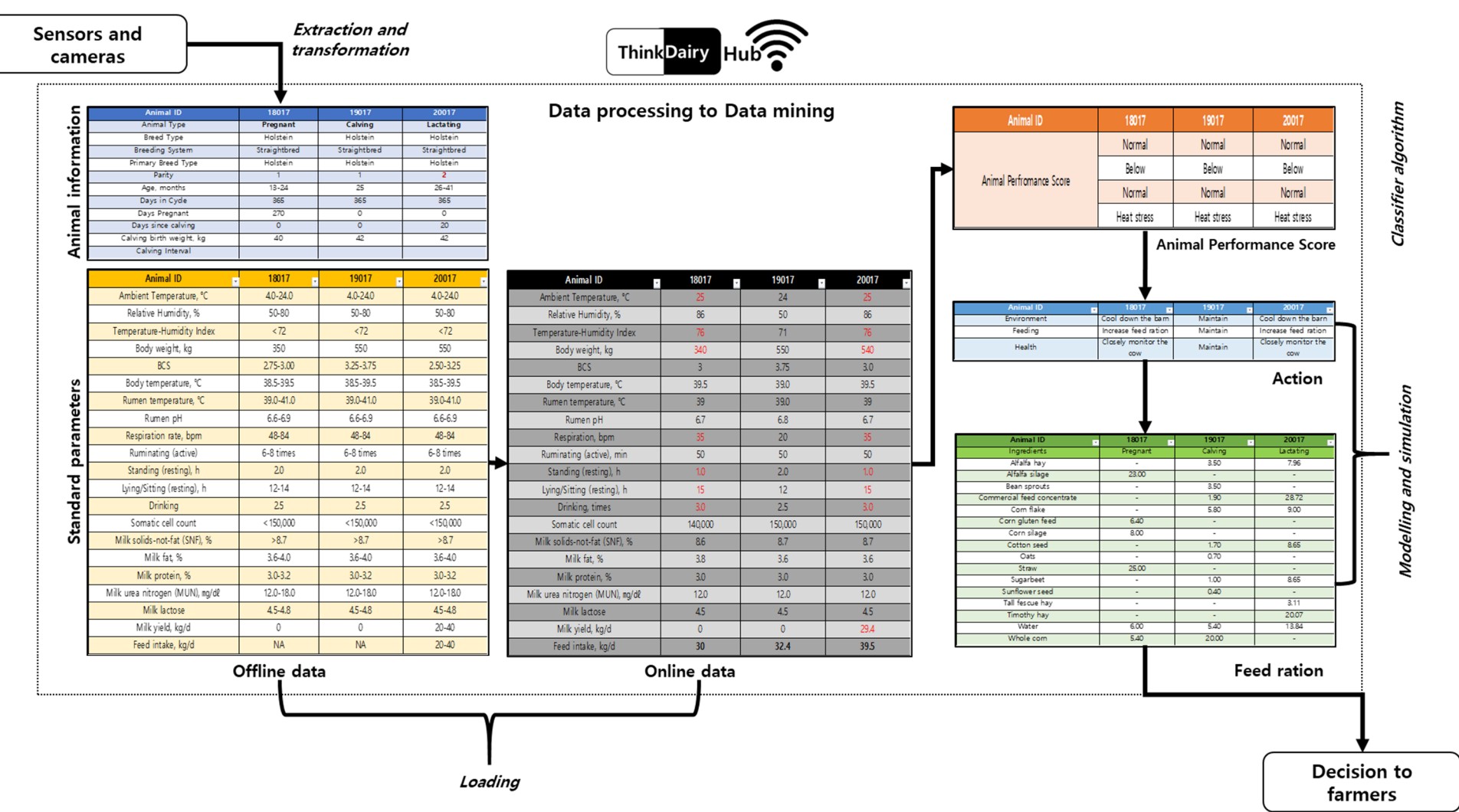

**Figure 2.** Theoretical Framework of the proposed ThinkDairy data-mining hub.

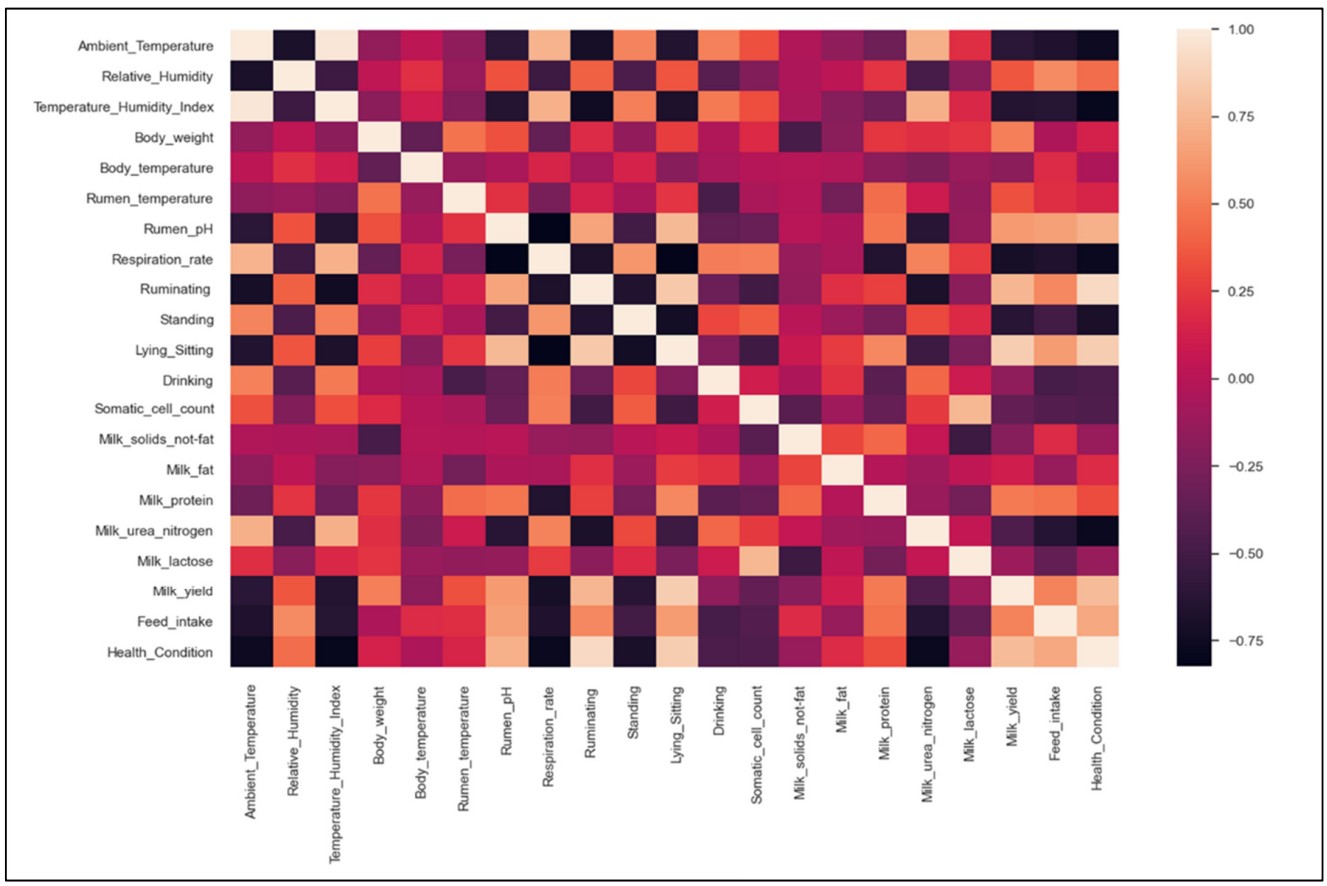

**Figure 3.** Pearson correlation of the parameters with the health condition of dairy cows.

### 3.3. Practical Application of the Data-Mining Hub

The overall accuracy and Kappa coefficient obtained by KNN in detecting heat stress and subclinical mastitis based on three different parameters are shown in Table 2. The highest overall classification accuracy in detecting heat stress and subclinical mastitis was observed in the group with the highest data parameters used (HI: 21) at 94.5% and 93.2% accuracy, respectively. This finding signifies that the data-mining hub that uses all the selected parameters identified and discussed in the above section could accurately detect heat stress and subclinical mastitis in dairy cows. The overall classification accuracy of heat stress in MID:13 and LO:8 groups were 88.6% and 84.8%, respectively, and 87.9% and 83.7% for subclinical mastitis, respectively. The MID:13 group, which includes the environmental condition, body temperature, standing, and lying behaviors, milk yield, and feed intake, could detect heat stress and subclinical mastitis at a moderately high accuracy level. The LO:8 group, which consisted of the environmental condition, milk yield, and feed intake parameters, resulted in moderately high accuracy detection of heat stress (84.8%), signifying that detecting heat stress in dairy cows could be based mainly on a few numbers of parameters. However, using a high number of parameters will result in the highest accuracy in detecting heat stress in dairy cows. The overall accuracy of detecting heat stress and subclinical mastitis increased with the increasing number of data parameters, signifying that increased use of information could result in more accurate detection of health conditions in dairy cows. The Kappa coefficient of HI:21 and MID:13 groups showed complete consistency with the true values of the data (>0.81), while the LO:8 group showed a high degree of consistency (0.79–0.80).

**Table 2.** The overall accuracy and Kappa coefficient of using the data-mining hub with a different number of parameters: HI (21 parameters), MID (13 parameters), and LO (8 parameters).

| Data Parameters | Heat Stress | | Subclinical Mastitis | |
|---|---|---|---|---|
| | Accuracy, % | Kappa Coefficient | Accuracy, % | Kappa Coefficient |
| HI (21) | 94.5 | 0.92 | 93.2 | 0.90 |
| MID (13) | 88.6 | 0.83 | 87.9 | 0.86 |
| LO (8) | 84.8 | 0.80 | 83.7 | 0.79 |

In order to study the effect of classification, the KNN algorithm was also used to obtain the precision of classifying the condition of the cows, as shown in Table 3. Among the three groups, only the data-mining hub using a high number of parameters (HI:21) reach more than 90% precision. The precision of recognition for the normal condition was at 92.8% and 93.7% for heat stress conditions in the HI:221 group, which reflected that the data-mining hub using the high number of data parameters could hardly misclassify the two different conditions. The same observation was noted in the effect of classification in detecting subclinical mastitis, normal condition, and others. The precision of recognition for normal condition and subclinical mastitis in the HI:21 group were 93.7% and 94.2%, respectively. This finding reflected that using the high number of data parameters will not easily misclassify the subclinical mastitis, normal, or other conditions of the dairy cows.

**Table 3.** The precision of the effect of classification using the data-mining hub with a different number of parameters: HI (21 parameters), MID (13 parameters), and LO (8 parameters).

| Data parameters | Precision, % | | Precision, % | | |
|---|---|---|---|---|---|
| | Normal | Heat Stress | Normal | Subclinical Mastitis | Others |
| HI (21) | 92.8 | 93.7 | 93.7 | 94.2 | 89.8 |
| MID (13) | 85.7 | 86.7 | 82.3 | 81.8 | 82.2 |
| LO (8) | 79.9 | 80.2 | 78.5 | 80.1 | 79.1 |

When the data-mining hub was used to detect normal and heat stress conditions in the same datasets of the dairy cows, the HI:21 group detected that 43.75% of the cows were heat-stressed, while in the MID:13 and LO:8 groups, 59.38% and 71.88% were detected as heat-stressed cows, as shown in Figure 4. The result of the practical application of the data-mining hub signifies that using a low number of parameters in detecting heat stress in dairy cows could overly detect heat stress conditions in dairy cows. The high percentage detection of heat-stressed cows (71.88%) using the LO:8 group could mean that the data-mining hub with the lower number of parameters over-detected dairy cows that are in normal condition and incorrectly classify them as being in a heat-stressed state. As evidenced in the precision of classification, the LO:8 group showed an 80.2% precision, indicating that there is an almost 20% percentage of misclassifying the dairy cows in heat stress. In the same way, a high percentage detection of dairy cows with risk of subclinical mastitis (50.0%) was observed in the data-mining hub using a lower number of data parameters (LO:8), while using a data-mining hub with a higher number of data parameters (HI:21) detected only 18.75%. This finding signifies that using a lower number of data parameters in the data-mining hub overly detect dairy cows being in a subclinical mastitis risk state, which are in fact in normal condition, as verified by the 80.1% precision of the LO:8 group. It is worthy to note that the MID:13 group detected 59.38% heat stress and 28.13% subclinical mastitis in dairy cows. Based on the overall accuracy and precision of classification, the MID:13 group showed moderately high accuracy and fairly high precision of classifying heat stress and subclinical mastitis in dairy cows. It should be noted that the combined effects of 13 parameters could reasonably classify and detect heat stress and subclinical mastitis in dairy cows, although 21 parameters still showed the highest accuracy and precision.

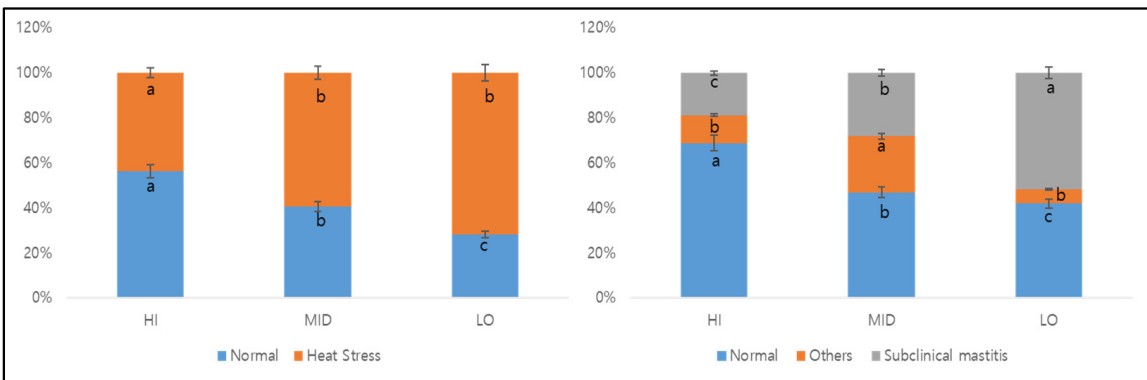

**Figure 4.** Percentage detection of normal, heat stress, and subclinical mastitis using the data-mining hub with the different number of parameters: HI (21 parameters), MID (13 parameters), and LO (8 parameters).

The precision of classification and overall accuracy of detecting heat stress and subclinical mastitis increased with the increasing number of data parameters used, signifying that the combined information such as the environmental condition, physiological and behavioral state of the cows, milk performance indexes, and feed intake could improve the precision of classification of conditions and accuracy of detection in dairy cows. The combination of the 21 parameters in detecting heat stress and subclinical mastitis in dairy cows improved the precision of classification and the accuracy of detection as compared to the data-mining hub that used only 13 and 8 parameters.

## 4. Conclusions

The challenges in adopting the integrated data support system are vast and multifactorial, which include the complex utilization of the massive flow of different and non-uniform data. In this study, it was demonstrated that a data-mining hub that includes only relevant information for detecting health conditions was explored. The development of the data-mining hub (ThinkDairy Hub) in this study clearly and concisely defined the target health condition: heat stress and subclinical mastitis to increase the accuracy of the detection. This study has developed a simple but innovative approach (data-mining hub that uses a simple and efficient classifying algorithm: KNN) in which combined information from different independent data was used to accurately detect the health and wellness status of the dairy cows. The practical application of the data-mining hub has shown that using 21 parameters instead of 13 and 8 parameters resulted in the high overall accuracy of detecting heat stress and subclinical mastitis in dairy cows with a high precision effect, reflecting the low percentage of misclassifying the conditions of the dairy cows. Using increased relevant information and combining them using the data-mining hub showed a potential AI-based tool to support decision making in dairy farming. For future studies, different classifying algorithms such as random forest, support vector machine, and artificial neural networks for the identification of the real-time condition of dairy, and modeling and simulation to arrive at the action and feed rations of the ThinkDairy data-mining hub, should be studied further to prove its application at the farm level as an AI-based tool in accurately analyzing the conditions of cows for precision feeding of the animals. Further studies on the different indicators for different health conditions, such as lameness and acidosis of dairy cows, should also be explored using the data-mining hub. In this article, it can be implied that an artificial intelligence-based tool, such as the proposed theoretical data-mining hub of dairy cows, could maximize the use of continuously generated but underutilized data in farms, thus simplifying the repetitive and difficult decision making tasks in dairy farming.

**Author Contributions:** Concept by J.P., D.-H.L. and S.H.C.; formal analysis by K.M.C.N., S.-i.C., S.S., M.-J.P., J.-P.C. and H.K.H.; methodology by K.M.C.N., S.-i.C., S.S., M.-J.P., J.-P.C. and H.K.H.; validation by D.-H.L., S.H.C.; writing of the original draft by K.M.C.N., J.P. writing: review and editing by D.-H.L., S.H.C. All authors have read and agreed to the published version of the manuscript.

**Funding:** This research was supported by the National Research Foundation (grant number NRF-2018R1D1A3B07048219) and Brain Korea 21 Center for Bio-Health Industry in Korea.

**Institutional Review Board Statement:** Not applicable.

**Informed Consent Statement:** Not applicable.

**Data Availability Statement:** The datasets from this study are available from the corresponding authors upon reasonable request.

**Conflicts of Interest:** The authors have no potential personal or financial conflict of interest relevant to this article to declare.

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
