# Peer review of "Precision Detection of Real-Time Conditions of Dairy Cows Using an Advanced Artificial Intelligence Hub"

_applsci, doi:10.3390/app112412043_

Round 1
Reviewer 1 Report
Authors conducted this research in the title of "Precision Detection of Real-Time Conditions of Dairy Cows using an Advanced Artificial Intelligence Hub".
The paper’s subject could be interesting for readers of journal. Therefore, I recommend this paper for publication in this journal but before that, I have a few comments on the text that should be addressed before publication:
Comments:
1)Materials and Methods: In this section authors used this word “our”. Words like “We”, “Me”, “Our” or “Us” are not common in article writing. Other words could be used by the authors. For example, this sentence "our work presented a new model in this work" could be replaced by this "A new model is presented in this work".
2) Figure 1: The used table in this figure is not sorted as table and has no related title. This issue should be addressed by the authors of this article. It is more integrated in terms of table and figure sorting. Also, the size of colored cells in this table could be smaller than what it is now. If it is changed, the whole size of this table could be smaller in size. Further more, the title and table could be on same page. This looks less misleading and confusing for readers of this work.
3) In Abstract section authors did not mention the main goal of this research. In other words, there is no obvious words about the main question that this research is managed to answer it in this section. If it is added, it would be really helpful for readers to enter and understand main purpose of this research. Furthermore, this would be useful for readers of this article to compare this work with similar works conducted in recent years.
5) Why authors did not used criteria like RMSE or MSE to evaluate the accuracy of data modelling in this work?. RMSE (Root-mean-square deviation) and MSE (Mean squared error) are achieved based on comparison between existed values and predicted values. These criteria could be useful for better evaluating of modelling accuracy in this article. It gives more clear vision to readers of this paper. In addition, MAE (Mean absolute error) criteria as extra criteria to assess accuracy of the modelling in this work.
6) Conclusion: In the conclusion section authors should mention more words about their suggestions to future works. It really can be helpful for future studies and works related with title of this article. For example, authors can mention some words about data base, software, number of indicators and etc.
7) Which software has been used in this work to modelling data and export the results?. Also, which software has been used in this work to export the diagrams in this work?. For example, software like MATLAB could be used for modelling and software like SigmaPlot could be utilized to export charts and diagrams.
8) What is new in this research in comparison with other similar works? In other words, authors should write more about novelty of this research. They should talk more about their innovations in this article. For example, authors should answer this question "Is the used model in this work unprecedented and completely new?". This would be useful for readers to compare this article with other similar works conducted in recent years.
9) This title "Selection of parameters to be used in the data-mining hub" in page 5 of this article should be more bold. This could be helpful for the reader to not confuse this title with before and after lines.
10) Figure 2: The used text fonts in tables of this figure are too small and unclear. These fonts could be more bold. It is really too hard to investigate this figure and get presented information in the figure.
11)Since recently it has been proved that artificial intelligence (AI) and machine learning has a numerous applications in all of engineering fields, I highly recommend the authors to add some references in this manuscript in this regard. It would be useful for the readers of journal to get familiar with the application of AI in other engineering fields. I recommend the others to add all the following references, which are the newest references in this field
[1] Roshani, M., Sattari, M.A., Ali, P.J.M., Roshani, G.H., Nazemi, B., Corniani, E. and Nazemi, E., 2020. Application of GMDH neural network technique to improve measuring precision of a simplified photon attenuation based two-phase flowmeter. Flow Measurement and Instrumentation, 75, p.101804.
[2] M. Forecasting house prices in Iran using GMDH. Int. J. Hous. Mark. Anal. 2021, 14, 555–568.
Reviewer 2 Report
The study is innovative and can help to solve important issues in livestock, relying on the large amount of data currently generated.
However, the study should be completed and validated with more farms and animals.
Regarding the general content, the first three sections of "Results and Discussion" should be included in "Materials and Methods", since the results in my opinion are only the section "Practical application of the data-mining hub". In addition, these results should be complemented by further discussion, based on the contribution of the different parameters used to the results and the accuracy of the tool. They should also discuss how it could be used at a practical level in livestock farming, and how the model could be adapted to other scenarios: other countries and even other species.
In particular, I would ask you to consider the following observations:
Featured Application:
Lane 13: replace the acronym AI with artificial intelligence (AI)
Keywords:
Lane 29: Review the words precision and accuracy, and the separation signs (;) and replace the acronym AI with artificial intelligence (AI)
Materials and Methods:
Lanes 80-81: Further describe the type and functionality of the sensors.
Lanes 104-108: We talk about the number of parameters, but they have not yet been described. They are not described until Table 1 of the "Results and Discussion" section, and should be described under "Materials and Methods". There is no description of how the different parameters are measured, and what reliability each would have.
Lane 138: Figure 1 is not sufficiently explained in the text.
Results and Discussion:
Lane 196: Table X by Table 1
Lane 240, 251 (and others): Revise the bibliographic citations in the text, in particular replace that of Armstrong, 1994 by (18).
Lane 276-289 (and others): Review the bibliography on "Milk performance indexes", in particular citation number 45, which seems very localistic. Include more literature to support the trend of parameters linked to milk quality.
Lane 300: Figure 2: This figure must be larger so that its content can be read well. Complete the "offline data" box with ranges or trends in all its parameters.
Lane 302: "Practical application of the data-mining hub": Improve the discussion of results.
Lane 376: Figure 3: Your format should be centered with your title.
Round 2
Reviewer 1 Report
All the comments have been addressed correctly
Reviewer 2 Report
Dear authors, I am pleased with your corrections on my comments, which have been made to help you better understand your article, and encourage you to continue this interesting line of research that can contribute to the improvement of the livestock sector.